# User Experience in VR Fashion Product Shopping: Focusing on Tangible Interactions

Jongsun Kim [1] and Jisoo Ha [1,2,*]

1   Department of Textiles, Merchandising and Fashion Design, Seoul National University, Seoul 08826, Korea; mstruck7@snu.ac.kr
2   Research Institute of Human Ecology, Seoul National University, Seoul 08826, Korea
*   Correspondence: jisooha@snu.ac.kr; Tel.: +82-3-880-1453

**Abstract:** It is necessary to focus on a VR environment centered on a tangible interaction (TI), which provides and interacts with the user experience (UX) with various sensational forms. Therefore, this study attempted to present UX evaluation items for VR fashion product shopping environment through focusing on the TI. In addition, the developed factors were evaluated for validity through empirical experiments and attempted to explore the possibility of using a VR shopping UX evaluation methodology. As a result of factor analysis of items related to VR shopping, six factors were extracted, and each factor was named as intention to use, playfulness, sharpness, telepresence, interactivity, and usability, respectively. As a result of a t-test for the difference in UX between immersive VR and non-immersive VR, it was found that there were significant differences in intention to use, playfulness, sharpness, and telepresence. As a result of performing a multiple regression analysis to analyze the effect of TI on the VR fashion shopping UX, it was found that interaction, playfulness, and telepresence are UX factors that are formed under the influence of TI in an immersive VR. In addition, intention to use, sharpness, telepresence, and usability were found to be factors experienced as an effect of TI in a non-immersive VR.

**Keywords:** VR fashion shopping; user experience; tangible interactions; immersive VR; non-immersive VR

## 1. Introduction

With the development of the Internet over the last 20 years, online shopping has made a leap forward, and owing to the recent COVID-19 pandemic, the purchasing of products and services through online platforms has become a common practice. Therefore, a new type of shopping environment based on augmented reality (AR), virtual reality (VR), and mixed reality (MR) environments is expected to become more prevalent [1]. This is an innovation in the distribution industry that started with the recent fourth Industrial Revolution and is expected to become smarter and more innovative with further developments of VR, AR, artificial intelligence (AI), and Internet of Things (IoT) [2,3]. In particular, the AR/VR field has been recognized by Information and Communication Technology (ICT) companies as a new future growth engine, with large-scale investments, and under such a new distribution environment, the focus is on the "experience" of the "humans".

Interest in an immersive or realistic interface that delivers vividness and sense of reality to users close to those of the real world is increasing. With the recent commercialization of fifth generation (5G) mobile networks, digital content can be consumed at a high speed without interruption through a high-performance network [4]. In addition, visual or auditory information is combined with digital content to further enrich the user experience and stimulate the five senses, thus providing a spatial experience similar to that of reality [5]. In this way, the immersive interface expands the user's cognitive system, including their senses, to make the virtual world feel similar to the real world. As user experience converges with digital technology, experiential access to products and services

through realistic interfaces is increasing. In addition, the proposed tangible interaction service simultaneously enables spatial and visual experiences similar to those of the real world through an immersive interface beyond the existing fragmentary and flat information delivery method [6,7]. However, despite this importance, many companies are still indiscriminately building VR/AR shopping services without applying most of the advantages of VR/AR and are only using them for one-time marketing purposes. There is a need to build a shopping environment that emphasizes the advantages of VR/AR, which are the ability to experience more realistic content than the online shopping environment and making the experience of shopping easy and fun for users [3]. Experiences with realistic products and services are important factors in shopping for apparel products [8,9]. The absence of a realistic tactile element, which has been pointed out as an important drawback of online shopping [10], can be a solution in terms of providing a realistic experience, which is an advantage of VR/AR. A VR/AR environment will enable multi-sensory and tangible interaction of fashion product shopping services, delivering more realistic and accurate product and shopping service information to users and enabling a realistic shopping experience. Furthermore, offline fashion shopping has changed the aim of offline stores in recent years, gradually shifting from the classic shopping concept of product searches and purchases in a way that conveys various in-store experiences and emotions of consumers. A number of studies have revealed that these new offline store experiences of consumers have a positive effect on brand recognition and evaluation [10–14]. From this perspective, a multi-sensory approach to product information that can supplement the shortcomings of online shopping or the provisioning of creative consumer experiences provided by offline stores is a significant advantage that can be provided through VR fashion shopping.

Recent application of VR/AR technology by fashion companies can convey the image of an innovative and creative brand, and thus, it is being used strategically by many different fashion brands [15]. In particular, it is worth noting the use of VR by global fashion companies; Italian luxury brands such as Dolce & Gabbana have implemented VR boutiques around the world, such as in Rome, Melbourne, Osaka, Miami, and Shanghai. Prada and Dior are also operating VR boutiques. In Korea, the Fendi Department Store has also implemented VR. These VR stores offer immersive VR environments, in which objects can be viewed while wearing the HMD, and a non-immersive environment, in which objects can be viewed with a smartphone or computer monitor. Research on the type of experience delivered to consumers when shopping for fashion products through VR, strategically provided by these various brands, and how consumers perceive the difference between immersive/non-immersive VR shopping experiences has yet to be systematically conducted. In addition, most of the prior related studies [16–20] have technically approached VR content implementation and need to be analyzed in terms of actual user experience.

User experience refers to the user's overall experience with products and services and allows them to clearly differentiate themselves from other products or services on the market, beyond simply usability or functions issues [21,22]. To understand the user experience more clearly, it is necessary to develop an appropriate evaluation tool and methodology. In particular, there is a need for a methodology that can analyze and evaluate the user experience [23] of a product or service experienced in a VR environment. VR content that moves the shopping space virtually allows users to experience fashion products, and because the purchase process is virtually implemented, user experience, such as the sense of presence during the shopping experience, is important. This indicates that to build user-centered products and services, it is necessary to evaluate the user experience (UX) in consideration of the characteristics of VR fashion product shopping. In this respect, it can be stated that the existing UX evaluation method that does not target the process of experiencing and purchasing fashion products is difficult to apply as a virtual fashion product shopping UX evaluation method. Therefore, in this study, a UX evaluation method suitable for virtual fashion product shopping based on the existing UX evaluation method has been proposed. This study aims to understand the retail

environment, which is changing into a convergence approach, and to examine the user experience that is developing multidimensionally in VR environment. In addition, it provides a user experience with various sensational forms and elements, attempts to understand the characteristics of the VR environment centering on tangible interactive interactions, and thus attempts to understand the evolving flow of user experience through technology and convergence. An additional purpose is examining the possibility of user experience in a VR environment that will become more active in the future. For that purpose, the structure of this research is as follows (Figure 1).

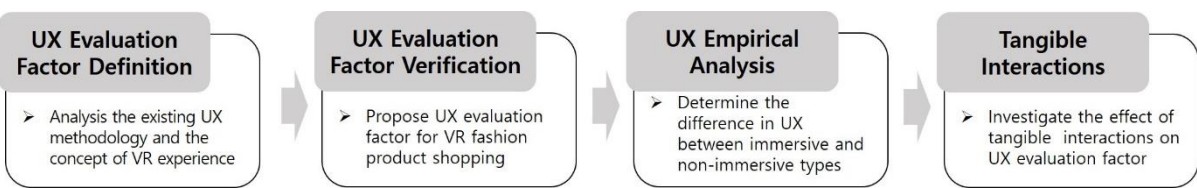

**Figure 1.** Research flow.

Section 2 describes the UX of a VR environment, the difference between immersive and non-immersive VR environments, and the preceding research on a tangible interaction. Section 3 proposes UX evaluation items for virtual fashion product shopping, derived based on the existing UX evaluation methodology. Section 4 interprets the results of measuring the user experience using the VR STORE operated by D&G. Finally, Section 5 provides some concluding remarks.

## 2. Background

### 2.1. User Experience for Virtual Reality

User experience (User eXperience, UX) is an extremely complex problem that includes the usage conditions, human emotions, and anticipation. Various factors affect the user experience caused by products or services [24,25]. In various fields, studies have been conducted on what UX is and what types of techniques can be used to collect its contents. Regarding the understanding of UX, the approach used in the design field mainly focuses on the interaction between the product (service), which is the object of a design, a person who uses a product, and the resulting experience. This approach is physical and makes it possible to explore product or service experiences of almost any dimension, such as cognitive, emotional, and aesthetic aspects, as well as the sensory aspect [26–28]. In particular, the UX of products or services implemented in a multidisciplinary manner can embrace a variety of perspectives in common, because various approaches may exist based on the unique knowledge system and methodology used in each academic field. It is therefore necessary to discuss the existing UX frameworks.

Based on previous studies related to a UX evaluation, Hiltunen et al. [29] explained which factors of products and services act as determinants of the overall user experience in the following five categories: utility, which recognizes the value of the providers and users through experience; usability to easily use a product or service; availability for easy use in various environments or situations; aesthetics, which induces interest in a service; and offline issues, covering other service support and business processes. Morville [30] presented the seven factors of useful, usable, desirable, valuable, findable, accessible, and credible as factors. In addition, Park [2] presented playfulness, usability, playful characteristics, aesthetics, and intention to use on a user experience evaluation scale, whereas Olsson et al. [31] categorized UX for mobile AR into 12 elements: captivation, collectivity, connectedness, creativity, efficiency and accomplishment, empowerment, increased awareness and knowledge, inspiration, intuitiveness, liveliness, playfulness and entertainment, and surprise. Dirin et al. [32] emphasized emotional engagement and explained it as adjustability, delightfulness, reliability, and satisfaction, as summarized in Table 1.

**Table 1.** User experience evaluation factors.

| UX factors | Hiltunen et al. (2002) | Morville (2004) | Park (2012) | Olsson (2013) | Dirin et al. (2017) |
|---|---|---|---|---|---|
| Utility | Utility | Useful | Utility | Empowerment Increased awareness and knowledge | Adjustability |
| Usability | Usability | Usable Findable | Usability | Efficiency and accomplishment | Reliability |
| Playfulness | - | Desirable | Playful characteristics | Playfulness and entertainment Surprise | Delightfulness |
| Aesthetics | Aesthetics | Valuable | Aesthetics | Captivation Creativity Intuitiveness Liveliness | - |
| Intention to use | Availability | - | Intention to use | Inspiration | Satisfaction |
| Social relations | Offline Issue | Credible | - | Collectivity Connectedness | - |
| Disability considerations | - | Accessible | - | - | - |

The UX factors they suggested can be summarized as utility, usability, playfulness, aesthetics, intention to use, social relations, and disability considerations. VR shopping is not a service type that multiple people experience at the same time, unlike VR games or sports. Therefore, among factors, according to the characteristics of product and service experience through VR shopping, social relations factor was excluded. Furthermore, since it is not a service specifically for people with disabilities, disability considerations factor was excluded as well. Therefore, five factors, utility, usability, playfulness, aesthetics, and intention to use, were proposed as UX evaluation factors for VR fashion product shopping.

Existing UX evaluation theory alone cannot accurately evaluate the specificity of immersive interfaces (immersive or realistic interface), such as AR, VR, and MR; the sense of presence, which refers to "the feeling of a real experience", and an immersive interface, which means the medium that conveys information, is the most realistic representation of the real world. Alternatively, such an interface can be defined as one in which all information is integrated and delivered to maximize the delivery of various types of elements that can provide a sense of presence and immersion, while overcoming the constraints of time and space in a virtual environment [33]. User senses have recently been expanded using sight, hearing, and touch (haptic), and the multidimensional immersive interface overcomes the constraints of space and time, making it possible to realize a vivid sense of realism based on various types of elements. As a next-generation medium that seeks to reproduce the real world in the closest way possible, high-quality expressiveness, sharpness, and a sense of presence have been provided. In particular, along with the medium, information is contained that allows users to feel with their five senses, such as wind, scent, tactile sensibility, and movement, and a change into a service that provides a new user experience through user interaction is occurring.

Looking at previous studies on user experience measurement factors in a VR environment, Shedroff [34] categorized them into sensory experience (interface), operational experience (interaction), and exploratory experience (information). In addition, Kim [35] categorized them into usability, emotional aspect, user values, and sense of reality while designing the UX for a VR indoor bike. Hur et al. [36] used the concepts of vividness and interactivity, along with usability, usefulness, playfulness, and intention to use. Vividness and interactivity are concepts that have been proposed by Steuer [37] as the characteristics of interfaces that provide VR experiences, and they have been used as major variables that affect the users' virtual experiences. Hong and Han [15] used immersion, emotional

value, functional value, suitability, willingness to recommend, willingness to experience the future, and willingness to purchase, whereas Jang and Chun [21] used the sense of presence, immersion, and interaction as explaining variables.

Telepresence indicates that the experience of the virtual environment or mediated environment feels like a part of reality and that the user feels as if an object that exists in that environment is physically real. In other words, it can be stated that the sense of presence is felt more in a mediated environment than in a physical environment [38]. Immersion refers to the feeling of being surrounded by virtual reality in an optimal state of experience, completely immersed in an activity that dominates the attention and perception system of the person [39]. Vividness is a concept related to how vividly information is delivered to the user's senses, and it can be considered a concept that explains the expressive richness of the environment as delivered by the VR interface. Interactivity is a concept related to how much a user can control a product or service [40]. Higher interactivity in a VR environment indicates that more users experience fun and enjoyment, for example, by experiencing a realistic shopping store [41].

Therefore, in this study, based on the characteristics provided by immersive interface, three factors of sense of presence, immersion, and sharpness were added to five factors of utility, usability, playfulness, aesthetics, and intention to use, which are important concepts of the VR experience, derived from the existing UX evaluation methodology. In addition, the UX evaluation factors for shopping for VR fashion products were proposed in Table 2. The developed factors were evaluated for validity through empirical experiments, and the possibility of using a VR shopping UX evaluation methodology was investigated.

**Table 2.** Survey factors for each item.

| Factor | Item | Abbreviation |
|--------|------|:------------:|
| Utility | By using VR, I was able to complete my shopping. | U1 |
| | Using VR allowed me to get the effect or results I expected to complete my shopping. | U2 |
| | I think VR is useful to get the shopping done. | U3 |
| | Using VR helped me complete my shopping. | U4 |
| Usability | I could easily perform what I was trying to do in VR shopping. | Us1 |
| | It didn't take much mental effort to use VR shopping. | Us2 |
| | I think VR shopping is easy to use. | Us3 |
| | How to use VR shopping was clear and easy to understand. | Us4 |
| Playfulness | VR shopping is as familiar as it has seen a lot. | P1 |
| | VR shopping is new and innovative. | P2 |
| | VR shopping feels live and dynamic. | P3 |
| | VR shopping is impressive and touching. | P4 |
| | VR shopping is creative and novel. | P5 |
| Aesthetics | The colors used in VR shopping are nice and attractive. | A1 |
| | The screen presented seemed to captivate all my senses. | A2 |
| | The layout of VR shopping is interesting and fun | A3 |
| Sense of presence | I felt like I was in a real shopping mall. | SP1 |
| | The visible scene felt like a real shopping mall | SP2 |
| | It is easy to recognize what the atmosphere of the real shopping mall is. | SP3 |
| | I felt like I could touch the products in the shopping mall. | SP4 |
| | I felt real objects embodied in a VR shopping mall. | SP5 |
| Immersion | The presented screen seems to have captured all my senses. | I1 |
| | The presented screen made me focus. | I2 |
| | I feel like I'm in a situation on the screen. | I3 |
| | It seems to be immersed in the situation. | I4 |
| | I felt the immersion on the presented screen. | I5 |

**Table 2.** *Cont.*

| Factor | Item | Abbreviation |
|---|---|---|
| Sharpness | The objects on the screen felt as if they were real. | SH1 |
| | Physical properties of space were perceived as reality. | SH2 |
| | The visible scenes felt realistic. | SH3 |
| | The movement in the screen was felt realistically. | SH4 |
| | The movement in the screen feels natural. | SH5 |
| Intention to use | If possible, I want to do fashion product shopping again through VR. | IU1 |
| | I will introduce VR fashion product shopping to my friends. | IU2 |
| | In a similar situation, I'd rather use VR fashion product shopping. | IU3 |
| | If there is a chance, I think I will use VR fashion product shopping again. | IU4 |

### 2.2. VR Environment: Immersive and Non-Immersive

VR refers to a computer environment that allows users to experience immersion in the virtual world generated by one or more types of virtual sensation synthesis feedback, such as sight and hearing [42]. In an artificially created world of senses, it refers to a cyberspace where users can feel a sense of presence and immersion and feel that they are actually in that space. Virtual reality is divided into non-immersive and immersive types, according to the method of experience. Non-immersive VR experiences the virtual world through an image output device, such as a monitor or projector, without wearing a separate device. Non-immersive virtual reality, using a monitor to allow users to easily experience virtual reality, is widely applied in 3D games and simulations. In early studies on virtual reality, non-immersive virtual reality using a monitor was often used under the title of desktop virtual reality [43].

Immersive virtual reality is a technology in which a user wears and experiences a head-mounted display (HMD) connected to a computer or mobile device [44]. Immersive virtual reality and non-immersive virtual reality can be clearly distinguished in terms of visuals. By wearing a separate device in an immersive virtual reality, the field of view of the user is controlled, the visual stimulus of the real world is blocked, and the user will only receive a 3D image output from the HMD [45,46]. In this way, it becomes possible to experience a virtual world as if it is the real world, and an intense sense of reality and immersion occur. Therefore, it can be expected that immersive virtual reality will make the users experience a higher sense of presence and immersion than non-immersive virtual reality, which will have a positive effect on a user experience evaluation.

However, in the case of immersive virtual reality, some users experience VR sickness and complain of dizziness, headaches, or nausea, among other issues [19], which hinder the sense of immersion or presence and will have a negative impact on a user experience evaluation in a real VR shopping environment [47]. In addition, the need to purchase special VR devices such as HMDs can negatively affect the intention to use and convenience in using VR fashion product shopping. Although an immersive VR environment simply provides a higher sense of presence and immersion, it is difficult to predict whether it can provide an improved user experience in a VR shopping environment compared to a non-immersive type. It is therefore necessary to compare immersive and non-immersive user experiences and discuss a method for providing virtual reality, which is advantageous for application to a real virtual VR shopping environment.

### 2.3. Tangible Interaction

An interface is an interaction method between humans and media, and it is not simply a tool to operate machines but is related to the way humans acquire and process information, communicate socially, and experience and understand the world. According to Heim [48], various interfaces have changed toward the formation of intersensoriality and



have changed from a visual interface expressed in letters, numbers, graphics, and images into a multiple sensory method that includes auditory and tactile sensations [49]. In particular, input/output interfaces, such as operation buttons, pointer devices, and vibrations, depend on tactile sensation, allowing users to experience an enhanced sense of control. It creates a tangible sensation by applying a spatial interaction with the visual images shown in the display through the tactile sensation interface. It is believed that by acquiring information from sight, exchanging audio information from tactile sensations, applying an input/output using tactile sensation, and finally, using hand and body movements as interfaces, a sense of presence in a VR environment can be achieved. Therefore, in a VR shopping environment, users will experience a multisensory interaction. As a type of multimedia appealing to multiple senses, this study examined the multisensory nature of a VR environment and tried to explain the way the human body actively experiences the world through VR when applying a tangible interaction [50].

Although a tangible interaction has been recognized as an important user interface element of a system, existing studies have mainly focused on the development of the system or interface itself. In response, Hornecker [51] created a system for tangible interactions, provided a conceptual framework, and emphasized its importance. Tangible interactions consist of four elements, and although each element does not exist as a completely mutually exclusive concept, each describes a tangible interaction from a different perspective. First, tangible manipulation refers to the physical interaction caused by the user's direct manipulation and is a concept related to whether the system supports such a manipulation. The second spatial interaction refers to the interaction given the meaning of space, such as movement and place, and it is related to how much the system allows the user to perceive and experience the meaning of the location or space. An expressive representation refers to interaction with an expressive power that converts physical objects into digital objects and is related to the level of expression of virtual objects described by the system. Finally, embodied facilitation refers to an interaction with a structure that promotes, allows, prohibits, or restricts the user's behavior. In particular, it is related to a physical structure that allows a large number of users to easily conduct a collective action. A tangible interaction can act as a factor improving the interaction between the system and user. Therefore, it can be expected that tangible interaction will act as a factor that improves the quality of interaction between the system and user, which is the core of the user experience in VR shopping environment. Tangible interaction factors in a VR fashion product shopping environment can be evaluated through the following three factors: tangible manipulation, related to experience in the direct shopping of fashion products; spatial interaction in terms of experiencing a real shopping space; and an expressive representation of physical objects, such as a product or store. The embodied facilitation element was excluded because the collective action (co-experience) formed through VR games or SNS is not a factor that can be experienced through shopping for VR fashion products.

## 3. Research Method

The UX experiment included understanding the target product for a UX evaluation through the experiment. In this study, a systematic experimental procedure was established to evaluate the shopping UX of VR fashion products. The procedure was largely composed of the development of a UX evaluation factor, UX experiment design, UX experiment performance, and derivation of the UX experiment results. Guiding this process, three research questions and hypotheses that focused this study were proposed as follows.

RQ1: What are the user experience evaluation measurement items for VR fashion shopping?

**H1.** *UX evaluation items for fashion product shopping consist of utility, usability, playfulness, aesthetics, intention to use, sense of presence, immersion, and sharpness.*

RQ2: What is the difference in UX evaluation between immersive and non-immersive VR types?

**H2.** *There will be differences between immersive and non-immersive VR in evaluating all UX items.*

RQ3: What are the tangible interactions that affect the user experience factors of fashion product shopping in a VR environment?

**H3.** *All tangible interaction factors will affect fashion product shopping UX in a VR environment.*

The detailed procedures of the experiment performed to solve these research problems are as follows.

### 3.1. UX Experiment Design

First, a VR store operated by D&G was selected as the experimental target to experience VR fashion product shopping. Compared to other brands, there were many VR boutiques that had been implemented and that provided systems that support both immersive and non-immersive environments. The immersive type was the D&G Cannes store, and the non-immersive type was the D&G Las Vegas store. Depending on the circumstances at the store, the composition of the product and the shape and structure of the store were slightly different, despite showing products of the same season. In addition, it was determined that the level of UX provided by each store would be similar in that the stores were implemented in major fashion cities in France and the United States. However, immersive and non-immersive store experiences were randomly assigned to remove bias that would affect the UX evaluation according to the order of such experiences according to the presence or absence of the HMD.

In the case of the immersive type, Oculus Quest 2, the latest HMD developed by Oculus, was used for visual experience, and for haptic experience, two wireless controllers were used to experience VR shopping. The non-immersive type looked at a 19-inch laptop monitor for visual experience, and for the haptic experience, VR shopping was conducted by clicking with a wireless mouse, moving to the store, and clicking on the products. For the auditory experience during each VR shopping experiment, the top-ranked audio source was provided by searching for luxury store music through a YouTube search on a smartphone. The immersive VR experiment was conducted in a state in which all external environments were visually blocked by wearing the HMD, and thus, the experiment environment was configured for free movements by removing obstacles within a radius of 2 m, while considering the movement of the body during the shopping process. Because the non-immersive VR experiment was presented through a laptop monitor, the experimental space was set against a gray wall without a pattern to block visual disturbances outside the monitor. The experimental environment was as follows (Figure 2).

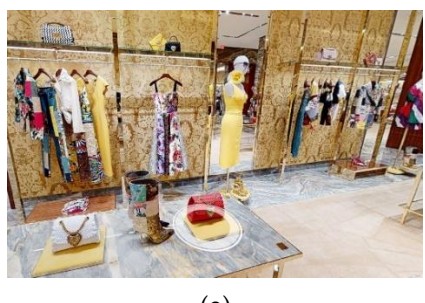  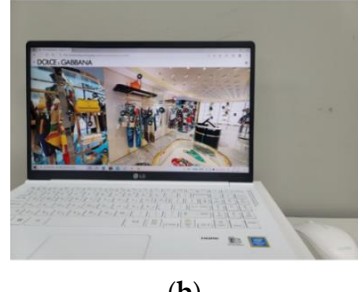

(**a**)  (**b**)

**Figure 2.** (**a**) Immersive VR type; (**b**) Non-Immersive VR type.

The participants were men and women in their 20s and 30s residing in the Republic of Korea. The detailed characteristic of participants of this study are presented in Table 3. The purpose of this study was to identify differences in user experience that appeared during the process of using recently emerged VR technology. Therefore, age was limited to control exogenous variables influencing the derivation of the results of the study. Therefore, it can be said that it was reasonable to select consumers in their 20s and 30s as research partici-

pants, because they have relatively high direct and indirect perceptions and experiences regarding new technologies. This research was approved by the SNU IRB for experiments. The approval number was 2104/003–011.

**Table 3.** Participants.

| | | N = 30 | | |
|---|---|---|---|---|
| | Classification | Frequency (Person) | Percentage (%) | Mean $\pm$ STDEV |
| Gender | Male | 7 | 23.3 | |
| | Female | 23 | 76.7 | |
| Age | 20's | 19 | 63.3 | 28.03 $\pm$ 4.85 |
| | 30's | 11 | 36.7 | |
| Education level | Undergraduate (attending) | 7 | 23.3 | |
| | Undergraduate (graduated) | 2 | 6.7 | |
| | Graduate (attending) | 14 | 46.7 | |
| | Graduate (graduated) | 7 | 23.3 | |
| Career | Housewife | 1 | 3.3 | |
| | Student | 19 | 63.3 | |
| | General office clerk | 6 | 20.0 | |
| | Professional occupation | 3 | 10.0 | |
| | Others | 1 | 3.3 | |
| VR (non-immersive) experience | No | 8 | 26.7 | |
| | Yes | 22 | 73.3 | |
| VR (immersive) experience | No | 11 | 36.7 | |
| | Yes | 19 | 63.3 | |

As a result of checking the general characteristics of the participants through a frequency analysis, in the case of gender, there were seven men (23.3%) and 23 women (76.7%); in the case of age, the average age was 28.03 years (standard deviation = 4.85); 19 individuals were in their 20s (63.3%), and 11 individuals were in their 30s (36.7%). In terms of academic background, 14 were attending graduates (46.7%), 7 were attending undergraduates (23.3%), 2 were graduated undergraduates, and 7 were graduated graduates (6.7%). There were 19 students (63.3%), 6 general office clerks (20.0%), 3 professionals (10.0%), 1 full-time housewife (3.3%), and 1 other (3.3%). In addition, 8 (26.7%) had no VR non-immersive experience, 22 (73.3%) had no VR immersive experience, 11 (36.7%) had VR non-immersive experience, and 19 (63.3%) had no VR immersive experience.

### 3.2. Experiment Procedure of UX

A total of 30 people participated in the experiment for four days from 3 May to 7 May 2021, and they were randomly assigned to the non-immersive and immersive VR experience experiments. After shopping for VR fashion products randomly selected among immersive and non-immersive types, they filled out a questionnaire, and another experiment was conducted. This experiment lasted approximately 40 min per person, and all experiments were normally terminated without complaints of motion sickness or headaches. When shopping for immersive VR fashion products, after becoming accustomed to the immersive VR environment by wearing the HMD and learning how to operate it, the participants shopped at the D&G VR store. After they indicated that they had shopped for a sufficient amount of time, they removed the worn devices and completed a questionnaire. The non-immersive type was conducted without wearing a separate device or without any special time limit, and similar to the immersive type, a questionnaire was filled out after the shopping ended. In both cases, background music for luxury stores was available from the starting point of the experiment.

*3.3. Derivation of UX Experiment Results*

The data from the UX experiment were analyzed using SPSS 26.0, and a factor analysis of the varimax rotation method was conducted to derive the UX evaluation factors. As the derived UX evaluation factors were obtained, the differences between immersive and non-immersive VR fashion-shopping experiences were analyzed. Correspondence t-test was conducted to analyze the difference in UX evaluation between the corresponding immersive and non-immersive types, and the average value was compared through a technical statistical analysis. To investigate the effect of tangible manipulation in VR fashion shopping, three factors of tangible manipulation were set, and the suitability of the evaluation factors was confirmed through a factor analysis and a reliability analysis of each item. Next, through a multiple regression analysis, the effects of tangible interaction factors on the shopping UX of VR fashion products were analyzed for both immersive and non-immersive types, the results of which are as follows.

## 4. Results and Discussion

*4.1. Factorial Validity and Reliability of Scale for User Experience Evaluation Measurement Factors for VR Fashion Shopping*

To explain RQ1, a factor analysis was performed on the items set for UX evaluation for VR fashion shopping. As a result of the reliability analysis of the UX evaluation, the overall Cronbach's value was 0.926, indicating that the reliability was extremely high, and for all sub-factors, it was 0.7, indicating that the reliability was also high. A factor analysis of the varimax rotation was conducted to define the user evaluation factors in the VR fashion product shopping environment, and the factor structure of the evaluation scale items based on the factor analysis is as follows (Table 4).

**Table 4.** Result of factor analysis for UX evaluation of VR fashion shopping.

| Item | Factors | | | | | |
|---|---|---|---|---|---|---|
| | Factor 1 | Factor 2 | Factor 3 | Factor 4 | Factor 5 | Factor 6 |
| IU4 | 0.829 | | | | | |
| IU1 | 0.823 | | | | | |
| IU3 | 0.812 | | | | | |
| IU2 | 0.801 | | | | | |
| U3 | 0.725 | | | | | |
| P5 | | 0.852 | | | | |
| P2 | | 0.849 | | | | |
| P4 | | 0.837 | | | | |
| P3 | | 0.706 | | | | |
| A1 | | 0.584 | | | | |
| A2 | | 0.441 | | | | |
| SH4 | | | 0.857 | | | |
| SH5 | | | 0.777 | | | |
| SH2 | | | 0.613 | | | |
| SH3 | | | 0.584 | | | |
| P1 | | | 0.423 | | | |
| SP5 | | | | 0.824 | | |
| SH1 | | | | 0.600 | | |
| SP3 | | | | 0.599 | | |
| SP4 | | | | 0.580 | | |
| U1 | | | | | 0.866 | |
| U4 | | | | | 0.790 | |
| U2 | | | | | 0.593 | |
| Us1 | | | | | 0.489 | |

**Table 4.** *Cont.*

| Item | Factors | | | | | |
|---|---|---|---|---|---|---|
| | **Factor 1** | **Factor 2** | **Factor 3** | **Factor 4** | **Factor 5** | **Factor 6** |
| Us4 | | | | | | 0.842 |
| Us2 | | | | | | 0.833 |
| Us3 | | | | | | 0.671 |
| Eigenvalues | 5.013 | 4.182 | 3.203 | 2.951 | 2.731 | 2.718 |
| Common variance (%) | 18.567 | 15.490 | 11.863 | 10.929 | 10.116 | 10.068 |
| Cumulative variance (%) | 18.567 | 34.057 | 45.920 | 56.848 | 66.946 | 77.032 |
| KMO = 0.803, Bartlett's $x^2$ = 1353.692, $p$ = 0.000 | | | | | | |

The KMO value was 0.803, which was more than 0.7, and the Bartlett test had a significance probability of 0.000, which was found to be appropriate at a significance level of 0.05. There were six items of the UX evaluation after removing the following items: A2, A3, I2, I3, I4, I5, P1 and P2. Therefore, H1, that the UX evaluation items for fashion product shopping would consist of utility, usability, playfulness, aesthetics, intention to use, sense of presence, immersion, and sharpness factors, was rejected. For the six factors, factor 1 accounted for 18.567% of the variance, factor 2 for 15.490%, factor 3 for 11.863%, factor 4 for 10.929%, factor 5 for 66.946%, and factor 6 for 10.068%. The cumulative variance % was 77.032%, which showed high explanatory power. Therefore, item 1 was intention to use, item 2 was playfulness, item 3 was sharpness, item 4 was telepresence, item 5 was interactivity, and item 6 was usability, and the UX evaluation factor in VR fashion product shopping was defined.

Hornecker [51] found the results of an item factor analysis set as a type of tangible manipulation, spatial interaction, and expressive representation, which are tangible interaction items in a VR shopping environment set based on the concept of a tangible interaction. The result of the factor analysis is as follows (Table 5).

**Table 5.** Result of factor analysis for tangible interaction.

| Item | Factor | | |
|---|---|---|---|
| | **Expressive Representation** | **Spatial Interaction** | **Tangible Manipulation** |
| ER1: It depicts fashion products, shopping malls, and products well. | 0.888 | | |
| ER2: Real fashion stores and products are visually well expressed. | 0.807 | | |
| ER3: Fashion product shopping space and product expression are realistically expressed. | 0.768 | | |
| SI1: It seems to be moving to a place where you actually shop for fashion products. | | 0.833 | |
| SI2: It seems like you're actually walking around a fashion product shopping mall. | | 0.772 | |
| SI3: I moved in the fashion product shopping space like usual offline shopping. | | 0.624 | |
| TM1: It feels like you are really touching the products. | | | 0.805 |
| TM2: It provides a shopping-like experience with direct manipulation. | | | 0.737 |
| TM3: It seems to be manipulating the actual clothes. | | | 0.681 |
| Eigenvalues | 2.565 | 2.143 | 2.131 |
| Common variance (%) | 28.499 | 23.808 | 23.674 |
| Cumulative variance (%) | 28.499 | 52.306 | 75.981 |
| KMO = 0.785, Bartlett's $x^2$ = 304.781, $p$ = 0.000 | | | |

The KMO value was 0.785, which was more than 0.7, the Bartlett test's significance probability was 0.000, and the factor analysis result was found to be suitable at a significance level of 0.05. Item 1 explained 28.499% of the variance, Item 2 explained 23.808% of the variance, and Item 3 explained 23.674% of the variance. The cumulative variance was 75.981%, which showed a high explanatory power, indicating that all items explained a tangible manipulation, spatial interaction, and expressive representation.

### 4.2. Immersive and Non-Immersive UX Evaluation

To analyze the difference in UX evaluation between immersive and non-immersive types, the RQ2, the corresponding t-test was performed (Table 6). A paired t-test established that the null hypothesis $H_0$ was based on the UX of immersive VR and the UX of non-immersive VR being the same, and the difference could be analyzed with the rule that 'if the significance level is less than 5%, $H_0$ will be rejected'. Therefore, as a result, intention to use (t = 2.873, $p < 0.05$), playfulness (t = 6.184, $p < 0.05$), sharpness (t = 2.061, $p < 0.05$), and telepresence (t = 5.187, $p < 0.05$) rejected the null hypothesis and were found to be factors that made the difference between the UX of immersive and non-immersive VR. On the other hand, there were no significant differences in interactivity (t = 0.785, $p > 0.05$) and usability (t = $-0.341$, $p > 0.05$). Therefore, H2, based on the difference in the evaluation of all UX factors between immersive VR and non-immersive VR, was rejected. Comparing the means, the immersive type showed a higher UX evaluation than the non-immersive type in terms of intention to use, playfulness, sharpness, and telepresence (Figure 3).

**Table 6.** Results of t-test. * $p < 0.5$; ** $p < 0.05$; *** $p < 0.001$.

| Category | | Mean | STDV | $t$ | $p$-Value |
|---|---|---|---|---|---|
| Intension to Use | Immersive | 3.96 | 0.767 | 2.873 ** | 0.008 |
| | Non-Immersive | 3.42 | 1.043 | | |
| Playfulness | Immersive | 4.16 | 0.568 | 6.184 *** | 0.000 |
| | Non-Immersive | 3.21 | 0.891 | | |
| Sharpness | Immersive | 3.20 | 0.714 | 2.061 * | 0.048 |
| | Non-Immersive | 2.85 | 0.874 | | |
| Telepresence | Immersive | 4.08 | 0.737 | 5.187 *** | 0.000 |
| | Non-Immersive | 3.18 | 0.710 | | |
| Interactivity | Immersive | 3.48 | 0.788 | 0.785 | 0.439 |
| | Non-Immersive | 3.37 | 0.703 | | |
| Usability | Immersive | 3.74 | 0.925 | $-0.341$ | 0.736 |
| | Non-Immersive | 3.80 | 0.874 | | |

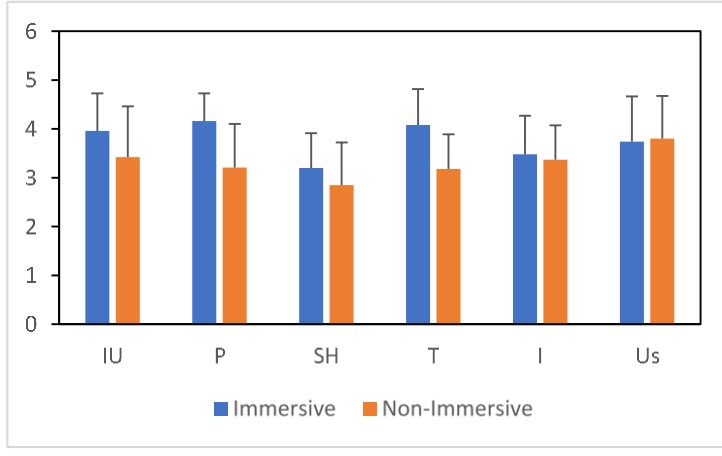

**Figure 3.** Comparing the means.

This means that the fashion product shopping experience in an immersive virtual reality environment makes users experience of intention to use, playfulness, sharpness, and telepresence stronger than the fashion product shopping experience in a non-immersive virtual reality environment. While shopping for VR fashion products, if the immersive method is used, information on the product or shopping space is felt more realistically than with a non-immersive type, and the user experiences the fun of VR shopping, because it is perceived as realistic. In this way, it can be determined that it has a positive effect on the intention to use VR shopping in purchasing fashion products.

Meanwhile, in terms of interactivity and usability, it was found that immersive and non-immersive types provide the same level of user experience. This means that there is no difference between the immersive and non-immersive types in terms of interactivity and ease of use, which is considered to be due to the difference between the presence or absence of the HMD and the individual's shopping purpose. Nelson [52] classified search goods and experiential goods based on the quantity or quality of information held by the consumers. Evaluating the characteristics of a product using only the information that a consumer has prior to purchasing a product is classified as a search goods evaluation, whereas for products that are difficult to evaluate before such direct experience, such evaluation is classified as an experiential goods evaluation; in addition, even with the same fashion product, the difference between shopping from the perspective of search goods or experiential goods may vary depending on the amount of personal information available and previous experiences. In the case of online shopping, because it is difficult to touch and experience goods, a search goods evaluation can be relatively more advantageous to consumers than an experience goods evaluation however, because VR-based shopping provides users with an offline shopping-like experience, it has the advantage of being able to indirectly experience products. VR shopping, using an immersive HMD wearing method, will make it possible to feel the clarity and genuineness of goods that are difficult to experience tangibly through an indirect experience in an immersive environment, as compared to the non-immersive type. Therefore, if the experience in the D&G VR store is searching for goods, the advantages of fast and convenient shopping will be pursued rather than factors such as store experience. In addition, it can be concluded that the non-immersive type, which can be controlled with a mouse without wearing the HMD, has a positive effect on user convenience and interaction. By contrast, in the case of experience goods, a direct experience of the product is important, and thus, it can be determined that the VR shopping experience when wearing an immersive HMD is positive for the usability and interactivity while shopping.

### 4.3. Effects of Tangible Interaction

A multiple regression analysis was performed to analyze the tangible interaction item affecting the UX factors of fashion product shopping in the VR environment, which was RQ3. The equation used for the multiple regression analysis was as follows (1). This explained the relationship between the three explanatory variables, X1, X2, and X3, TM, SI, and ER, and the six UX factor Yn dependent variables.

$$Y_n = b_0 + b_1 X_1 + b_2 X_2 + b_3 X_3 \tag{1}$$

First, the results of analyzing the influence of tangible interaction among fashion product shopping UX factors in an immersive VR environment were as follows. A multiple regression analysis was conducted to analyze the factors of a tangible interaction influencing the user experience of fashion product shopping in an immersive VR environment (Table 7).

**Table 7.** Results of multiple regression analysis for immersive VR. * $p < 0.5$; ** $p < 0.05$; *** $p < 0.001$.

| Dependent Variable | Independent Variable | Non-Normalized Coefficient | | Standardized Coefficient | T | R² (adj. R²) | F |
|---|---|---|---|---|---|---|---|
| | | B | Standard Error | β | | | |
| Intention to use | (constant) | 0.169 | 0.964 | | 0.175 | 0.421 | 6.294 ** |
| | TM | 0.211 | 0.161 | 0.226 | 1.308 | | |
| | SI | 0.261 | 0.209 | 0.235 | 1.248 | (0.354) | |
| | ER | 0.477 | 0.275 | 0.330 | 1.735 | | |
| Playfulness | (constant) | 1.396 | 0.753 | | 1.855 | 0.356 | 4.796 ** |
| | TM | 0.085 | 0.126 | 0.123 | 0.676 | | |
| | SI | 0.070 | 0.163 | 0.086 | 0.432 | (0.282) | |
| | ER | 0.507 | 0.215 | 0.473 | 2.361 * | | |
| Sharpness | (constant) | 0.344 | 0.963 | | 0.357 | 0.334 | 4.343 * |
| | TM | 0.193 | 0.161 | 0.222 | 1.198 | | |
| | SI | 0.339 | 0.209 | 0.329 | 1.628 | (.257) | |
| | ER | 0.203 | 0.275 | 0.151 | 0.741 | | |
| Telepresence | (constant) | −0.169 | 0.722 | | −0.234 | 0.649 | 15.992 *** |
| | TM | 0.335 | 0.121 | 0.374 | 2.777 * | | |
| | SI | 0.324 | 0.156 | 0.304 | 2.069 * | (0.608) | |
| | ER | 0.435 | 0.206 | 0.313 | 2.114 * | | |
| Interactivity | (constant) | −1.168 | 0.675 | | −1.730 | 0.730 | 23.477 *** |
| | TM | 0.279 | 0.113 | 0.291 | 2.468 * | | |
| | SI | 0.603 | 0.146 | 0.530 | 4.120 *** | (0.699) | |
| | ER | 0.310 | 0.193 | 0.209 | 1.610 | | |
| Usability | (constant) | 0.394 | 1.365 | | 0.288 | 0.202 | 2.195 |
| | TM | −0.181 | 0.228 | −0.162 | −0.795 | | |
| | SI | 0.169 | 0.296 | 0.126 | 0.571 | (0.110) | |
| | ER | 0.737 | 0.389 | 0.423 | 1.894 | | |

Tangible manipulation (B = 0.279, $p < 0.05$) and spatial interaction (B = 0.603, $p < 0.001$) had a significant effect on interactivity. The explanatory power was 73.0%, and the regression model was found to be suitable (F = 23.477, $p < 0.001$). The tangible interaction had an effect on playfulness and telepresence, and the expressive representation (B = 0.507, $p < 0.05$) had a significant effect on playfulness, and the explanatory power was 35.6%. It was found that tangible interaction (B = 0.335, $p < 0.05$), spatial interaction (B = 0.324, $p < 0.05$), and expressive representation (B = 0.435, $p < 0.05$) all had a significant effect on telepresence. The explanatory power was 64.9%, and the regression model was found to be a suitable model (F = 15.992, $p < 0.001$). Tangible interaction did not significantly affect intention to use, sharpness, and usability. Therefore, H3, based on all tangible interaction factors (TM, SI, ER) that will affect the UX of fashion product shopping in the VR environment, was rejected.

As factors influencing telepresence, it was confirmed that tangible manipulation, spatial interaction, and an expressive representation are all related, which can be judged as experiencing telepresence in an immersive VR shopping environment through complex sensory interaction experiences. Interaction experiences can be interpreted as being influenced by factors of spatial interaction and tangible manipulations formed by hand and body movements rather than visual-oriented expressive representations. In terms of playfulness, only an expressive representation was found to be an influencing factor, which can be judged as the realistic visual experience in an immersive HMD-wearing VR environment, being a factor that induces the entertainment experience.

The results of analyzing the effect of a tangible interaction that affects the UX during fashion product shopping in a non-immersive VR environment are as follows (Table 8).

**Table 8.** Results of multiple regression analysis for non-immersive VR. * $p < 0.5$; ** $p < 0.05$; *** $p < 0.001$.

| Dependent Variable | Independent Variable | Non-normalized Coefficient | | Standardized Coefficient | $t$ | $R^2$ (adj. $R^2$) | F |
|---|---|---|---|---|---|---|---|
| | | B | Standard Error | B | | | |
| Intention to use | (constant) | −0.778 | 0.628 | | −1.239 | 0.677 | 18.164 *** |
| | TM | 0.469 | 0.155 | 0.412 | 3.018 ** | | |
| | SI | −0.141 | 0.164 | −0.126 | −0.858 | (0.640) | |
| | ER | 0.869 | 0.193 | 0.630 | 4.507 *** | | |
| Playfulness | (constant) | 0.835 | 0.746 | | 1.120 | 0.375 | 5.206 ** |
| | TM | 0.173 | 0.185 | 0.178 | 0.938 | | |
| | SI | 0.368 | 0.195 | 0.385 | 1.881 | (0.303) | |
| | ER | 0.185 | 0.229 | 0.157 | 0.808 | | |
| Sharpness | (constant) | −0.464 | 0.621 | | −0.748 | 0.549 | 10.558 *** |
| | TM | 0.155 | 0.154 | 0.162 | 1.006 | | |
| | SI | 0.205 | 0.163 | 0.219 | 1.261 | (0.497) | |
| | ER | 0.565 | 0.191 | 0.489 | 2.960 ** | | |
| Telepresence | (constant) | 0.772 | 0.530 | | 1.458 | 0.503 | 8.788 *** |
| | TM | 0.094 | 0.131 | 0.122 | 0.718 | | |
| | SI | 0.295 | 0.139 | 0.388 | 2.126 * | (0.446) | |
| | ER | 0.305 | 0.163 | 0.324 | 1.872 | | |
| Interactivity | (constant) | 1.526 | 0.600 | | 2.541 | 0.350 | 4.667 * |
| | TM | 0.284 | 0.149 | 0.370 | 1.913 | | |
| | SI | 0.069 | 0.157 | 0.091 | 0.436 | (0.275) | |
| | ER | 0.225 | 0.185 | 0.242 | 1.221 | | |
| Usability | (constant) | 1.208 | 0.759 | | 1.592 | 0.326 | 4.198 * |
| | TM | 0.134 | 0.188 | 0.140 | 0.712 | | |
| | SI | −0.062 | 0.199 | −0.066 | −0.312 | (0.249) | |
| | ER | 0.615 | 0.233 | 0.532 | 2.636 * | | |

In the case of the non-immersive type, tangible manipulation (B = 0.469, $p < 0.001$) and expressive representation (B = 0.869, $p < 0.001$) were found to have a significant effect on intention to use, with an explanatory power of 67.7%. It was found that expressive representation (B = 0.565, $p < 0.001$) had a significant effect on sharpness, with an explanatory power of 54.9%. For telepresence, spatial interaction (B = 0.295, $p < 0.05$) had a significant effect, whereas expressive representation (B = 0.615, $p < 0.05$) had a significant effect on usability. The sub-factor of tangible interaction did not significantly affect playfulness or interactivity. Therefore, H3, based on all tangible interaction factors (TM, SI, ER) that will affect UX of fashion product shopping in the VR environment, was rejected.

In the case of the non-immersive type, intention to use experience was also an item which was affected by tangible manipulation and an expressive representation. It can be stated that this is because the VR environment is experienced through the control of the monitor screen and mouse; in addition, it is easy to move within a shopping space or manipulate the products, and the expression of products and stores implemented in VR can be a factor increasing the intention to use for VR shopping. In addition, it was confirmed that the expressive representation item, which provides a visual immersion experience, plays an important role in sharpness. In the case of telepresence, tangible manipulation, spatial interaction, and expressive representation all affected compound sensory interaction in the immersive type, whereas in the case of the immersive type, it was found that the experience of controlling the space, such as moving the actual shopping mall, was a more important factor than visual immersion. In terms of usability, only an expressive representation among tangible interactions was found to have an effect, and it can be stated that visually experiencing actual fashion products and stores acted as a convenient factor in VR fashion product shopping.

## 5. Conclusions

The experiences of products or services based on realistic interfaces that trick the user's senses to make a virtual world feel as if it is the real world are increasing. As user experience converges with digital technology, experiential access to products and services through realistic interfaces is also increasing. In addition, it is necessary to focus on a VR environment centered on tangible interactions, which provides and interacts with the user experience with various sensational forms and elements beyond the fragmentary and flat information delivery method of a conventional medium. Therefore, this study attempted to present user experience evaluation items in a VR fashion product shopping environment, which was limited in explanation by the existing user experience concepts and theories, by focusing on tangible interactions. In addition, the developed factors were evaluated for validity through empirical experiments and attempted to explore the possibility of using a VR shopping UX evaluation methodology. To this end, UX evaluation items of virtual fashion product shopping derived based on the existing UX evaluation methodology were proposed, and the user experience was measured using the VR STORE currently operated by D&G. This experiment distinguished between non-immersive virtual reality, which is the easiest way for users to experience virtual reality using a monitor, and immersive virtual reality, which allows users to experience a strong sense of reality and immersion by accepting only 3D images output from the HMD. The influence of a tangible interaction during the process was also analyzed.

By analyzing previous studies related to UX evaluation, five items (utility, usability, playfulness, aesthetic, and intention to use) related to the experience characteristics of products and services through VR shopping including were selected. In addition, three factors, sense of presence, immersion, and sharpness, were added to reflect the specificity of the VR interface. The items developed in this way were subjected to factor analysis and reliability analysis using SPSS 26.0, and the suitability of the UX evaluation items was verified. As a result of the reliability analysis of the UX evaluation, the overall Cronbach's value was 0.926, indicating very high reliability, and all sub-factors showed high reliability, with Cronbach's values of 0.7 or higher. As a result of factor analysis, six factors were extracted, and the cumulative variance was 77.032%, which showed high explanatory power. Each factor was named as intention to use, playfulness, sharpness, telepresence, interactivity, and usability, respectively, and was defined as a user experience evaluation factor in VR fashion product shopping. The reliability and factor analysis results of the items set by tangible manipulation, spatial interaction, and expressive representation, which were tangible interaction items in the VR shopping environment set based on tangible interaction concept of Hornecker, were found to be suitable. The cumulative variance was 75.981%, which showed high explanatory power, suggesting that all items explained tangible manipulation, spatial interaction, and expressive representation well.

An empirical experiment was conducted based on the UX evaluation items developed in this way. In order to analyze the difference in UX between immersive virtual reality and non-immersive virtual reality, as a result of testing, it was found that there were significant differences in intention to use, playfulness, sharpness, and telepresence. Furthermore, comparing the averages, intention to use, playfulness, sharpness, and telepresence all showed that the immersive type had a higher UX evaluation than the non-immersive type. This means that the fashion product shopping experience in the immersive virtual reality environment allows users to experience intention to use, playfulness, sharpness, and telepresence stronger than the fashion product shopping experience in the non-immersive virtual reality environment. If an immersive method is used in the process of shopping for VR fashion products, the information of the products or the shopping space is more realistic than the non-immersive type, and it can be said that the pleasure of VR shopping can be experienced, and this positively affects the intention to use VR shopping to purchase fashion products.

As a result of performing a multiple regression analysis to analyze the effect of tangible interaction on the VR fashion product shopping user experience, it was found

that interactivity, playfulness, and telepresence are UX items that are formed under the influence of tangible interaction in an immersive VR environment. In addition, intention to use, sharpness, telepresence, and usability were found to be items experienced as an effect of tangible interaction in a non-immersive VR environment. In the case of the immersive type, tangible manipulation, spatial interaction, and expressive representation all affect multi-sensory interaction, whereas in the case of the non-immersive type, rather than visual immersion, the experience of controlling space, such as moving in an actual shopping mall, can be seen as a more important factor for presence. In immersive VR, playfulness and interactivity, and in non-immersive VR, intention to use, sharpness, and usability, were found to be affected by tangible interaction, which can be explained by the difference in the presence or absence of HMD. It can be said that in the immersive VR, it is possible to have fun and to have an interaction using multiple senses, because the virtual visual experience completely controlled through the HMD is overwhelming. On the other hand, in the non-immersive type, the monitor screen and mouse control are exposed to an interface similar to the existing online shopping environment. Therefore, the focus is on a more improved visual experience compared to the existing online shopping environments, and this can be seen as experiencing intention to use and usability, such as moving easily in the shopping space and getting help with manipulation for product search.

This study is meaningful in that it proposed UX evaluation items for VR fashion product shopping that were not theoretically established in previous studies and verified them through empirical experiments. UX evaluation in a VR or AR environment is meaningful in that the UX evaluating items, including sharpness, immersion, and sense of presence, were similar and used overlappingly, and they were applied as a vague notion, which was modified and improved. Shopping for fashion products is not only about price or convenience of shopping, and emotional experience is also an important factor. Therefore, the results of research conducted on shopping for fashion products can be applied to various retail areas. In particular, since the purchase of not only fashion products, but also all products are made online rather than in offline stores, the UX evaluation study conducted for VR fashion shopping is an important study that can provide important information for the practical and industrial application of VR technology.

In addition, it is meaningful to provide data necessary to build a suitable VR environment in various areas by comparatively analyzing UX in immersive and non-immersive VR environments through experiments. In particular, wearing the HMD may cause problems, such as cost and utility in building a realistic VR retail environment, and may even act as a hindrance to VR shopping for some individuals. Therefore, it will be possible to determine the implementation of an immersive and non-immersive environment for each UX item that is important to consumers in each product or service, which will greatly contribute to the establishment of a VR retail strategy. Finally, tangible interaction is increasingly important in UX, which changes from a visual interface expressed with letters, numbers, graphics, and images to a multisensory way that includes auditory or haptic. Therefore, by applying the concept of tangible interaction to VR shopping UX, evaluation items were developed, and it is meaningful that a pioneering study was conducted to analyze the detailed items of tangible interaction that form each factor. It is hoped that future research will be expanded to UX of VR for various products and services, and it is expected that research will be conducted to emphasize the importance of tangible interaction in the UX of VR.

**Author Contributions:** Conceptualization, J.K. and J.H.; methodology, J.K. and J.H.; formal analysis, J.K. and J.H.; investigation, J.K.; writing—original draft preparation, J.K.; writing—review and editing, J.H.; visualization, J.K.; supervision, J.H. All authors have read and agreed to the published version of the manuscript.

**Funding:** This research received no external funding.

**Institutional Review Board Statement:** The study was conducted according to the guidelines of the Declaration of Helsinki and approved by the Institutional Review Board of Seoul National University (2104/003–011).

**Informed Consent Statement:** Informed consent was obtained from all subjects involved in the study.

**Acknowledgments:** The authors would like to thank all participants of the user study and Mashhadi Abolghasem Fatemeh for proof-reading.

**Conflicts of Interest:** The authors declare no conflict of interest.

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
