# Peer review of "User Experience in VR Fashion Product Shopping: Focusing on Tangible Interactions"

_applsci, doi:10.3390/app11136170_

Round 1

Reviewer 1 Report

Table 1 is dated as the most recent article cited is Park (2012). For the last 10 years or so, much progress has been made in this field. Authors are urged to look at some of the latest publications.

Fig 2 - I don't think it is appropriate to show the D&G outlet photo in a learned journal. Scientific publication should be free of commercialism.

Fig3(a) does not mean much. Should show user in a VR scene rather just wearing some gears.

Overall, there is little science in the paper. All analyses are standard statistical treatment from user studies. The scientific content has to be improved. 

Reviewer 2 Report

The authors of this paper have presented an user experience evaluation in a virtual reality fashion product shopping environment. The paper is quite well structured, and clearly describes the problem addressed, but there are a few things to clarify.

First of all, you should clearly highlight what is new about the paper and what elements it brings in addition to the current scientific literature. Then, finally, this should be reinforced by referring to other similar works, by comparing the results with those.

I would also suggest adding some pictures from the experiment and a figure with the regression model from SPSS, containing the dependent, independent variables and their links (create your own Figure, do not use a PLS screenshot).

The English language and style require minor corrections, but it is good and clear to be understood.

You used the word ‘thesis’ at the end of the introductory chapter, probably referring to the article. Please check (line 108).

In Table 1 the word ‘playfulness’ should be written in capital letters.

The next sentence is very difficult to follow (lines 147-151): “Among them, 147 in relation to the characteristics of product and service experience through VR shopping, 148 social relations is excluded because VR shopping is not a service type that multiple people 149 experience at the same time, unlike VR games or sports and since it is not a service specif-150 ically for people with disabilities, disability considerations factor was also excluded.” Please break it down into several sentences to be easier to understand.

Chapter 3.2 is written twice (lines 339-353 are the same with lines 367-381).

The title of section 4.1 appear twice (line 383 and 491).

There are some typos that should be corrected, such as: “…absence of an HMD …” (line 438), repetition: “..five items were selected as items related..”.

Review reference 39.

Reviewer 3 Report

User Experience in VR Fashion Product Shopping: Focusing on Tangible Interactions

In this article authors attempted to present user experience evaluation items in a virtual reality fashion product shopping environment by focusing in tangible interaction using empirical experiments. Several factors were evaluated for validity through the study. From the result of factor analysis of items related virtual reality shopping, 6 factors were extracted, and each factor was named as intention to use, playfulness, sharpness, telepresence, interactivity, and usability. Furthermore, as a result of t-test for the difference in user experiences between immersive virtual reality and non-immersive virtual reality, they concluded that there were significant differences in intention to use, playfulness, sharpness, and telepresence.

+This report demonstrated that authors clearly understood and properly integrated the concepts and knowledge of the field. In addition, authors presented concluding remarks showing analysis and synthesis of ideas. 

+Overall, majority of components of a typical research report are clearly stated. However, needs minor modifications as mentioned below.

-Paragraph (lines 109-114) – Change word “Chapter” to “Section” for clarity reason.

-While majority of required components of a good and typical research report are presented, “Research Questions” are not clearly stated; in addition, it is technically appropriate (almost required) to list “Null Hypotheses” that later could be either accepted or rejected in the results section.

-Tables of results and analysis could be summarized (recommended, not required). Graphs could be included to allow a visual outlet for readers.

Round 2

Reviewer 2 Report

The authors have made a considerable work to revise their study. I propose the acceptance of their work.